# The Effect of Functional Ceramsite in a Moving Bed Biofilm Reactor and Its Ammonium Nitrogen Adsorption Mechanism

Liangkai Wang [1,2], Ningyuan Zhu [3,*], Hiba Shaghaleh [3], Xinyu Mao [1], Xiaohou Shao [1,2], Qilin Wang [1] and Yousef Alhaj Hamoud [4,*]

[1] College of Agricultural Science and Engineering, Hohai University, Nanjing 210098, China
[2] Jiangsu Province Engineering Research Center for Agricultural Soil-Water Efficient Utilization, Carbon Sequestration and Emission Reduction, Hohai University, Nanjing 210098, China
[3] College of Environment, Hohai University, Nanjing 210098, China
[4] College of Hydrology and Water Resources, Hohai University, Nanjing 210098, China
[*] Correspondence: zhuny0106@hhu.edu.cn (N.Z.); yousef-hamoud11@hotmail.com (Y.A.H.)

**Abstract:** For aquaculture wastewater with low ammonium nitrogen concentration, combining the carrier adsorption and biological nitrogen removal processes can maximize their respective advantages. Functional ceramsite that has excellent ammonium nitrogen adsorption performance and excellent biocompatibility was the key to the moving bed biofilm reactor (MBBR) adsorption—shortcut simultaneous nitrification and denitrification (shortcut SND) process. Our group prepared a high-strength lightweight ceramsite that met those requirements. In this study, we applied functional ceramsite in MBBR to cope with low-concentration ammonium aquaculture wastewater. The findings show that utilizing functional ceramsite as a filler was conducive to the adhesion of microorganisms. The biofilm has a minimal effect on the adsorption capacity of ceramsite due to the existence of pores on its surface. Our study further examined the $NH_4^+$-N adsorption mechanism of bio-ceramsite. The Freundlich adsorption isotherm model and the quasi-second-order kinetic model had better fitting effects on the $NH_4^+$-N adsorption process. The adsorption of bio-ceramsite to $NH_4^+$-N was an endothermic process that included physical and chemical adsorption. Furthermore, the results of adsorption thermodynamics suggested that bio-ceramsite has an affinity for the adsorption of $NH_4^+$-N. Consequently, this functional ceramsite can be a promising option for MBBR to improve nitrogen removal from aquaculture wastewater.

**Keywords:** moving bed biofilm reactor; ceramsite; shortcut simultaneous nitrification and denitrification; aquaculture wastewater; adsorption of ammonium

## 1. Introduction

Eutrophication is a severe problem that has worsened in recent years due to the discharge of ammonium nitrogen containing effluent from the intensive aquaculture industry [1]. The development of denitrification technology that can efficiently remove nitrogen while also being simple to use and energy-efficient is thus urgently needed. The trickling filter, biological contact oxidation process, moving bed biofilm reactor (MBBR), and biological aerated filter are some of the biofilm technologies that are increasingly being implemented in wastewater treatment [2–4]. The MBBR process has been proven to effectively address the issues with regular backwashing of biological aerated filters, the large occupation area of the rotating biological contactor, the complex operation of cleaning and replacing the filler in the biological contact oxidation tank, and the large energy consumption of the fluidized bed high-density carrier's fluidization [5–8]. The pollutant removal performance of the reactor is determined by the suspended filler in the MBBR, which provides adequate growth and reproduction conditions for microbial attachment.

Various biocarriers have been introduced in the MBBR process to date, including plastic media, polyurethane sponge, activated carbon, naturally occurring materials, non-woven carriers, ceramic carriers, etc. [9–11]. The multi-sided hollow ball and cylindrical suspension packing of polypropylene and polyethylene both have advantages in terms of strength and stability, and can be processed into various shapes [12]. However, they also have several drawbacks, including a poor hydrophilicity of fillers, the easy falling-off of biofilms, a lack of microporous structure, an inadequate biological microenvironment, and a limited capacity to adsorb ammonium nitrogen pollutants [13]. In contrast, lightweight ceramsite is inexpensive and simple to obtain, exhibiting a rough surface and a dense internal pore structure. It can form a more stable aerobic zone and anoxic zone/anaerobic zone in different parts of ceramsite, thereby creating a stable microenvironment for the simultaneous nitrification and denitrification (SND) process of the whole reactor [14]. It can function effectively as an alternative to MBBR plastic filler. Zhou et al. [15] treated urban domestic wastewater using suspended ceramics as the MBBR and achieved a high SND efficiency, with the average nitrification rate and the denitrification rate reaching 2.21 mg $NH_4^+$-N/(g MLSS·h) and 0.98 mg $NO_3^-$-N/(g MLSS·h), respectively. If the nitrification process in SND is controlled at the nitrite nitrogen stage to achieve shortcut SND denitrification, the alkali demand in the nitrification process, as well as the carbon source and oxygen supply required for denitrification, can be reduced, and the reaction time can be shortened to achieve a more efficient and energy-saving denitrification process [16]. Several approaches have been proposed to achieve long-term partial nitrification in activated sludge systems by controlling operation conditions, such as dissolved oxygen (DO), temperature, pH, free ammonia (FA), and free nitrous acid levels [17–19]. Additionally, the FA concentration in the reactor can be controlled to realize the nitrification process by adjusting the $NH_4^+$-N concentration, pH value, and temperature [19]. Zeolite, as an ammonium nitrogen adsorption material, can generate a higher concentration of FA between the zeolite surface and the liquid phase when the adsorption equilibrium is attained, thereby inhibiting nitrite-oxidizing bacteria on the biofilm surface. Chen et al. [20] achieved the shortcut biological nitrogen of low-concentration ammonium nitrogen wastewater through zeolite adsorption and biochemical desorption. In addition, since the concentration of ammonium nitrogen in aquaculture wastewater is low and the quality of effluent fluctuates greatly, combining the carrier adsorption and biological nitrogen removal processes can maximize their respective advantages, boost the efficiency of wastewater treatment, and optimize the entire process.

Miladinovic et al. [21] use natural zeolite as a filler material in the packed bed system with the goal of combining the $NH_4^+$-N selective adsorption of zeolite and the nitrification of microorganisms to synergize the removal of $NH_4^+$-N from wastewater. The results showed that the combined system outperformed simple microbial denitrification in that it could better adapt to the rapid change in ammonium nitrogen concentration. However, the ion exchange resin and zeolite often have densities higher than that of water (1.92~2.88 g/cm$^3$), which do not meet the requirements of MBBR for the lightweight carrier. Through preliminary research, we prepared a high-strength lightweight ceramsite with a high porosity and a large specific surface area using sediment, zeolite, and bentonite as raw materials, which not only exhibits excellent ammonium nitrogen adsorption performance, but is also suitable for the adhesion and growth of microorganisms, fulfilling the requirements of adsorption shortcut SND processes for carriers [22].

Based on functional ceramsite, the primary goals of this study were (1) to evaluate the biofilm culturing performance of functional ceramsite in the start-up stage of MBBR; (2) to examine the effects of environmental parameters on the $NH_4^+$-N adsorption of bio-ceramsite (when functional ceramsite is attached to the biofilm, it becomes bio-ceramsite), such as the biofilm, contact time, adsorbent dosage, and pH; and (3) to shed light on the adsorption mechanism of bio-ceramsite through isotherm and kinetic models.

## 2. Materials and Methods

### 2.1. Wastewater Composition

The synthetic wastewater was comparable to the water quality of the Baima Lake aquaculture pond. The $NH_4^+$-N concentration of the wastewater was about 26.2 mg/L and the chemical oxygen demand (COD) concentration was about 100 mg/L. The composition of synthetic wastewater is displayed in Table 1.

**Table 1.** Components of synthetic wastewater.

| Primary Nutrients | | Trace Elements | |
|---|---|---|---|
| **Components** | **Concentration (mg/L)** | **Components** | **Concentration (mg/L)** |
| $NH_4Cl$ | 100 | $CuSO_4$ | 0.005 |
| Glucose | 167 | $CoCl_2 \cdot 6H_2O$ | 0.021 |
| $NaHCO_3$ | 107 | $Na_2MoO_4 \cdot 2H_2O$ | 0.016 |
| $KH_2PO_4$ | 10.2 | $ZnSO_4 \cdot 7H_2O$ | 0.041 |
| $FeSO_4 \cdot 7H_2O$ | 2.51 | $H_3BO_3$ | 0.15 |
| $MgSO_4 \cdot 7H_2O$ | 5.71 | $MnCl_2 \cdot 4H_2O$ | 0.211 |
| $CaCl_2 \cdot 2H_2O$ | 1.54 | Vitamin D | 0.0002 |

### 2.2. Material Characteristics of Functional Ceramsite

Functional ceramsite using zeolite (clinoptilolite, with cation exchange capacity of 0.31 meq/g), dredging sediment, and bentonite as composite materials was developed based on our earlier findings [22]. The physico-chemical properties of the composite ceramsite are depicted in Table 2.

**Table 2.** The physical characteristics of functional ceramsite.

| Items | Functional Ceramsite | Criterion [1] |
|---|---|---|
| Sum of the Breaking rate and Wear rate (%) | 1.63 | $\leq 6.00$ |
| Silt carrying capacity (%) | 0.21 | $\leq 1.00$ |
| Solubility in hydrochloric acid (%) | 0.94 | $\leq 2.00$ |
| Void fraction (%) | 70.23 | $\geq 40.00$ |
| BET specific surface area ($m^2/g$) | 52.68 | $\geq 0.5$ |
| Apparent density ($g/cm^3$) | 1.62 | – |
| Bulk density ($g/cm^3$) | 0.78 | – |
| Compressive strength (MPa) | 2.3 | – |
| Porosity (%) | 53.93 | – |
| pHzpc | 4.43 | – |
| CEC | 99.87 | – |

[1] China's standard method of Artificial Ceramsite Filter Material for Water Treatment (CJ/T 299-2008).

### 2.3. Experimental Set-Up and Operation of MBBR

One pilot-scale MBBR with an effective volume of 6 L (diameter: 11 cm, height: 75 cm) is illustrated in Figure 1.

Suspended functional ceramsite was added to the experimental reactor and corresponded to a volume fraction of 20% ($V_{support}/V_{reactor}$). The simulated wastewater entered the upper part of the reactor and exited at the bottom of the reactor. The water inlet was connected with a peristaltic pump, and the hydraulic residence time of the system was adjusted by controlling the inlet flow. An aerator was placed at one side of the bottom of the tank, providing good oxygen transfer into the synthetic wastewater, and an airflow meter monitored the airflow rate. The agitator caused the ceramsite to circulate around the whole chamber. Moreover, water quality sensors were also equipped to monitor the change in water quality parameters, such as DO and pH in the reactor.

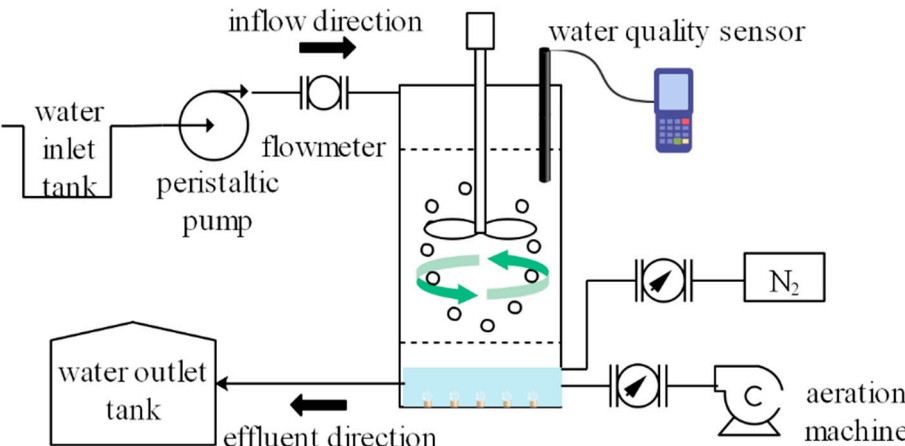

**Figure 1.** Schematic diagram of the experimental facility.

MBBR was initiated by introducing seed sludge obtained from the Nanjing City wastewater treatment plan. The inoculated sludge was discharged from the reactor when the microorganism of sludge attached well to the surface of carriers after 24 h. During the early stages of cultivating the biofilm, synthetic wastewater was added to the reactor in a sequencing-batch mode. The reactor ran on a 12 h cycle duration that included 40 min of water inflow, 9 h of aeration, 60 min of standing, and 80 min of water outflow. The DO concentration stabilized around 2.0–3.0 mg/L. The temperature was kept at 25 °C and the pH was regulated between 7.5 and 8.0. The initial biofilm appeared on the surface of functional ceramsite after 5 days of continuous operation, and the residual mud–water mixture in the reactor was completely emptied on the 6th day. Considering the instability of the biofilm at the initial stage, continuous inlet water with a smaller flow rate was used and continuous inlet water with a larger flow rate was employed as the biofilm on the carrier became thicker.

*2.4. Static Adsorption Experiments*

Solutions with $NH_4^+$-N concentrations of 1 mg/mL were prepared using ammonium chloride, based on which the simulated ammonium nitrogen wastewater with varying concentrations was configured. Each bio-ceramsite was treated with 2 M NaCl solution at 22 °C by shaking for a period of 24 h to remove adsorbed ammonium nitrogen, followed by a wash in distilled water to remove any excess NaCl on the surface.

2.4.1. Batch Adsorption Analysis

The batch adsorption experiments were conducted in 250 mL conical flasks at 20 °C to reveal the adsorption performance of ammonium nitrogen by bio-ceramsite. The effects of biofilms (i.e., fresh functional ceramsite or bio-ceramsite), the effects of different adsorbent dosages: (2, 4, 6, 8, 10, 16, 20, 24, 28, 32, 36, and 40 g/L, respectively), regulating the pH of the water between 4 and 11, and the initial $NH_4^+$-N concentration (10, 25, 50, 100, 150, 200, and 300 mg/L, respectively) were investigated. The flasks were positioned in a thermostatic shaker and agitated for 8 h at a fixed agitation speed of 120 revolutions per minute (rpm) at 20 ± 1 °C to reach a steady state $NH_4^+$-N concentration. At the end of the sorption experiment, the solution was filtered via a 0.22 μm filter membrane, and then the concentrations of ammonium nitrogen in the filtered solution were measured. Each experiment was triplicated to obtain the average value.

2.4.2. Adsorption Isotherms

A 2.5 g sample of bio-ceramsite was placed into conical flasks with 250 mL of $NH_4^+$-N solution. Varying $NH_4^+$-N concentrations ranging from 5 to 50 mg/L were added to the 250 mL solution (i.e., 5, 8, 10, 15, 25, 35, and 50 mg/L). The flasks were placed in a

thermostatic shaker and agitated for 8 h at a fixed agitation speed of 120 rpm at 5 °C, 20 °C, and 35 °C (278.15 K, 293.15 K, 308.15 K), respectively. The solution was filtered using a 0.22 μm filter membrane, and then the concentrations of ammonium nitrogen in the filtered solution were determined at the end of the sorption experiment. The experimental results were fitted by an adsorption isotherm and the adsorption thermodynamic state parameters could be calculated based on the $NH_4^+$-N adsorption data of bio-ceramsite.

### 2.4.3. Adsorption Kinetics

A 2.5 g sample of bio-ceramsite was placed in a 250 mL $NH_4^+$-N solution with an initial concentration of 25 mg/L. The pre-determined contact times for the experiments were 15, 30, 60, 90, 120, 180, 240, 300, 360, 420, and 480 min. The flasks were placed in a thermostatic shaker at a constant agitation speed of 120 rpm at 20 °C. At the end of the sorption experiment, the solution was filtered through a 0.22 μm filter membrane, and then the concentrations of ammonium nitrogen in the filtered solution were determined. The adsorption capacity of the bio-ceramsite per unit ($q_t$) was computed and modeled using the adsorption time (t).

### *2.5. Analysis Methods for Water Quality*

### 2.5.1. Water Quality Analysis Method

$NH_4^+$-N, $NO_2^-$-N, N—N, and total nitrogen (TN) of the sample were detected by the a UV/visible spectrophotometer (Shanghai Puyuan Instrument Co., LTD, Alpha-1860Plus, Shanghai, China) in accordance with the standards for the examination of water and wastewater [23]. The pH and DO were identified by a water quality analyzer (HACH Company, HQ-30d, Loveland, CO, USA).

### 2.5.2. Evaluation of Biomass and Biological Activity

The biomass of ceramsite biofilm was measured by the weighing method [24]. Specific oxygen uptake rate (SOUR) was used to characterize the biological activity of the biofilm (mg $O_2$/(L·h·mg VS)). SOUR is a measure of the microbial metabolic rate based on the OUR per unit of biomass, which has the advantages of being easy to calculate and being able to account for variations in macro microbial activity [25]. The specific measurement procedures included taking representative bio-ceramsite from the reactor, rinsing the fillers with simulated wastewater to eliminate any leftover impurities on the biofilm, and repeating the cleaning procedure three to four times. A 20 g sample of cleaned bio-ceramsite was added into a triangular flask with a stopper (250 mL), the DO saturated simulated wastewater was added to the bottle mouth, the DO meter was inserted into the bottle with a stopper, the change in DO with time every 1 min was recorded, and the DO-t curve was created. The formula for calculating SOUR was as follows:

$$\text{SOUR} = \frac{DO_1 - DO_2}{(t_2 - t_1) \times \text{VS}} \tag{1}$$

where $DO_1$ and $DO_2$ (mg/L) denote DO levels at $t_1$ and $t_2$, respectively, and VS (mg) denotes the volatile solid content of the biofilm.

### 2.5.3. Extraction and Measurement of EPS

Extracellular polymeric substances (EPS) are macromolecular substances that exist outside microbial cells in biofilms. Their chemical constituents are quite complicated, including polymers, such as extracellular polysaccharide (Po), extracellular protein (Pr), humic acid, deoxyribonucleic acid (DNA), and phospholipid [26]. Among them, Po and Pr account for the largest proportion and are thus the main components of EPS [27]. Previous studies have shown that heat extraction is the simplest method, the amount of EPS extracted by heat extraction is substantial, and the amount of DNA generated by cytolysis is the smallest among various methods [28]. In this test, the heating extraction method was selected for extracting EPS from biofilm: the representative fillers were taken from the

reactor and placed into a 50 mL centrifuge tube, and 0.9% NaCl solution was added to completely submerge the fillers. Additionally, the tube was heated in an 80 °C water bath before high-speed centrifugation (12,000× $g$ for 20 min) to obtain the supernatant and biofilm solid. The EPS solution will be prepared by using a 0.22 μm filter membrane to filter the supernatant, and the protein and polysaccharide concentrations were assessed using the coomassie brilliant blue method and the phenol-sulfuric acid method [29].

### 2.5.4. Analysis of Biofilm Morphology

Scanning electron microscopy (SEM, ZEISS MERLIN compact-61–78) was also used to characterize bio-ceramsite and functional ceramsite in order to determine the morphological characteristics of the biofilm on ceramsite. In addition, the biofilm morphological changes in the culturing process were observed by an optical microscope.

## 3. Results and Discussion

### 3.1. Membrane Hanging Start-Up Characteristics of Bio-Ceramsite

### 3.1.1. Changes in Various Nitrogen during MBBR Start-Up Period

When the original biofilm was formed on the surface of the ceramsite, the mud water mixture was completely emptied and the continuous water inflow began. Figure 2 depicts the changes in ammonium nitrogen and TN at MBBR start-up stage.

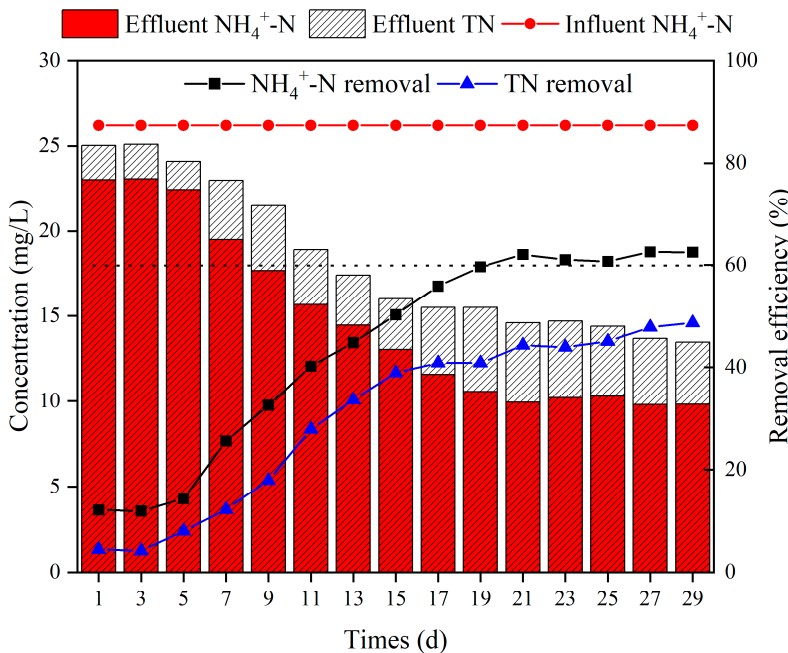

**Figure 2.** The removal efficiency of ammonium and TN during the start-up period of MBBR.

As shown in Figure 2, the concentration of $NH_4^+$-N in the reactor did not change considerably from day 1 to day 5, despite the continuous water inflow and biofilm culturing. This may be due to the long generation time of nitrifying bacteria, which was unable to adapt properly to the environment at the initial stage, and hence reduced the nitrification effect [30]. After microorganisms acclimated to the environment, the concentration of outlet water $NH_4^+$-N decreased rapidly from day 5 to day 21, while the ammonium nitrogen removal efficiency of the reactor stabilized at about 60% after day 21. The change pattern of TN was similar to that of ammonium nitrogen, and the removal efficiency was around 48% when it was stable. The reduction of TN in the reactor was, on the one hand, due to the assimilation of microorganisms in the process of biofilm culturing, and nitrogen was used to synthesize cell components in the biofilm; on the other hand, the rich pore structure inside the ceramsite created a favorable anaerobic microenvironment for the

attachment and growth of denitrifying bacteria, while the biofilm on the surface of the ceramsite was suitable for the growth of aerobic nitrifying bacteria, forming a nitrifying bacteria-denitrifying bacteria denitrification chain on bio-ceramsite [31]. The stabilization of ammonium nitrogen and TN removal efficiency were a crucial indicator of the successful start-up of the MBBR.

### 3.1.2. Change Rules of Biomass and Biological Activity

In this test, the biomass and biological activity of ceramsite in the reactor were continuously monitored following the continuous inflow and film culturing in the MBBR to determine whether the microorganisms on the ceramsite carrier biofilm had attained stable growth. Figure 3 demonstrates that the patterns of biofilm biomass and biological activity in the reactor were similar to each other, and could be divided into three stages. The first stage (day 1–day 5) was the adaptation stage for microorganism attachment and growth. The biofilm had little biomass and weak biological activity, which caused the effluent ammonium nitrogen concentration to be high. The second stage (day 5–day 21) was the exponential growth stage of microbial growth. On the one hand, the cumulative growth rate of biofilm biomass in this stage was as high as 1.12 mg VS/(g carrier·d), and reached 21.4 mg VS/(g carrier) on the 21st day; on the other hand, after the 13th day, the SOUR of microorganisms was stable at 2.89–3.08 mg $O_2$/(L·h·mg VS), which showed that the biofilm at this stage had a high biological activity, and the effluent concentrations of ammonium nitrogen and TN decreased significantly. The third stage (day 21–day 29) was the stabilization stage of microbial growth. The biomass loaded by bio-ceramsite was essentially stable at approximately 21.8 mg VS/(g carrier), whereas the biological activity of the biofilm decreased in the later stage of the reaction, which may be caused by the increase in the proportion of anaerobic denitrifying bacteria in the ceramsite and the decrease of oxygen consumption with the growth and thickening of the biofilm [32].

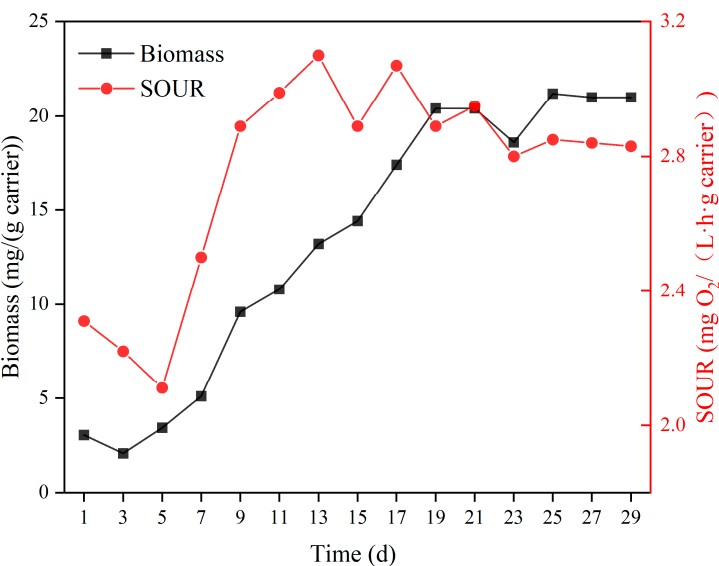

**Figure 3.** Variations of immobilized biomass and biological activity in the carrier.

### 3.1.3. Mechanism of Biofilm Cultivating

In this test, EPS was characterized by the relative content of Pr/VS and Po/VS. Figure 4 illustrates that the relative content of Pr and Po in the biofilm showed a phased change trend as the biofilm grew. The cell activity was low and the ability to secrete EPS was limited throughout the adaptation stage of microbial growth (day 1 to day 5). The relative content of Pr and Po decreased slightly at this stage. In the rapid growth stage (day 5 to day 21), the biological activity was restored, which facilitated the rapid secretion of EPS. The average growth rates of Pr/VS and Po/VS were 8.65 and 2.27 mg/(g·d), respectively;

by day 17, the total content of EPS reached its maximum value of 377.32 mg/g during the whole start-up process. The significant quantity of EPS firmly fixed microorganisms on ceramsite and promoted the stability and rapid maturation of the biofilm structure [33]. The ability of the biofilm to secrete EPS was essentially stable in the microbial stabilization stage (day 21–day 29), and Pr/VS and Po/VS were 285.67 ± 9.24 mg/g and 78.07 ± 4.49 mg/g, respectively. Numerous studies have revealed that the change in the Pr/Po ratio can serve as a characteristic index of biofilm growth and shedding, and the Pr/Po value of the biofilm in a stable reactor is around 3.38–4.57 [32]. After day 21, the value of Pr/Po in this test was stable at about 3.70, just between 3.38 and 4.57, indicating that the biofilm on the ceramsite was mature and the reactor's biofilm formation had been successfully initiated. From the above analysis, it can be seen that using functional ceramsite as filler is conducive to the adhesion and aggregation of microorganisms.

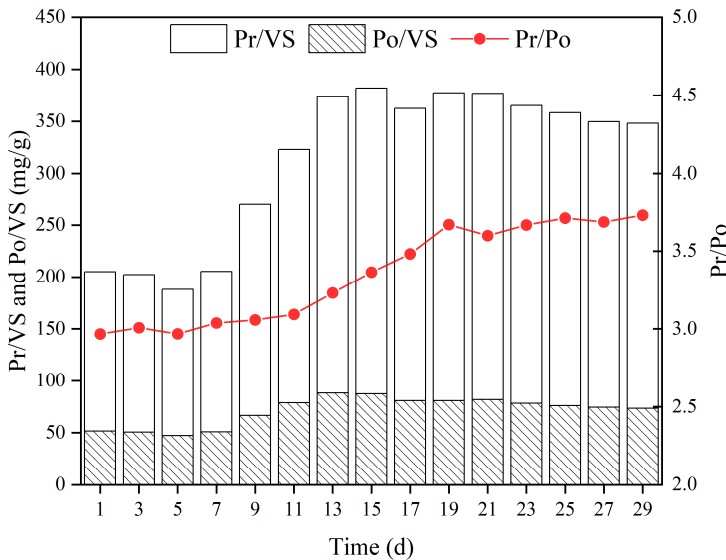

**Figure 4.** Variation of EPS during the start-up period of MBBR.

Taken together, the presence of EPS facilitated the attachment and aggregation of microorganisms on the surface of ceramsite. Feng et al. [34] suggested that electrostatic force inhibited microbial attachment if EPS was low at the initial culturing stage. As an essential component of biofilm EPS, Po played an important role in enhancing the tightness and stability of the biofilm structure, and was usually employed to quantify the strength of biofilm attachment ability. The relative content of Po remained at 73.5 ± 2.59 mg/g after the successful biofilm formation, suggesting that the biofilm exhibited strong attachment and stability within this polysaccharide range. The proportion of Pr in the biofilm predominated over that of Po, and these differences vary greatly across stages. Part of ammonium nitrogen in the wastewater were assimilated by microbes to make proteins, which lead to the increase in the protein proportion relative to polysaccharide and the reduce of ammonium nitrogen [35]. Shi et al. [36] believed that the change in protein content in EPS reflected the ability of microbial cells to secrete extracellular enzymes that were primarily involved in the capture and transportation of pollutants. In general, the secretion of protein is positively associated with the ammonium nitrogen removal efficiency of MBBR.

### 3.1.4. Analysis of Biofilm Morphology

Microscopic changes in the biofilm before and after the biofilm formation of ceramsite were observed, as displayed in Table 3. Ceramsite underwent a transition from loose to dense biofilm formation. The biofilm structure was relatively fragmented at the initial stage of biofilm culturing. The surface of ceramsite was substantially covered by the biofilm with a complete structure after 21 days of cultivation. It can be assumed that the ceramsite had

successfully completed the biofilm formation process at this time. The color of the biofilm on the surface of ceramsite was yellowish brown after biofilm formation, which was mainly composed of aerobic microorganisms; the biofilm inside the ceramsite pores was black gray, primarily made up of anaerobic microorganisms [37].

**Table 3.** Characteristics of the biofilm on bio-ceramsite.

| Time | Characteristics of Biofilm Culturing |
|---|---|
| 1 d | There was light yellow thin mucosa at the initial stage of biofilm formation. |
| 5 d | The biofilm was thin, incomplete, and unevenly distributed. |
| 13 d | The yellow mucous membrane on the surface thickened, and biofilm appeared on most of the surface of ceramsite. |
| 21 d | The surface of the filler was entirely covered by a smooth biofilm with a complete and dense structure. The biofilm surface was yellowish brown. |
| 29 d | The surface biofilm of ceramsite was yellowish brown, while the internal biofilm was black gray, and were inhabited by aerobic and anaerobic microorganisms, respectively. |

After 21 days of continuous culture, bio-ceramsite with successful film formation and functional ceramsite without film formation were analyzed by SEM, as shown in Figure 5.

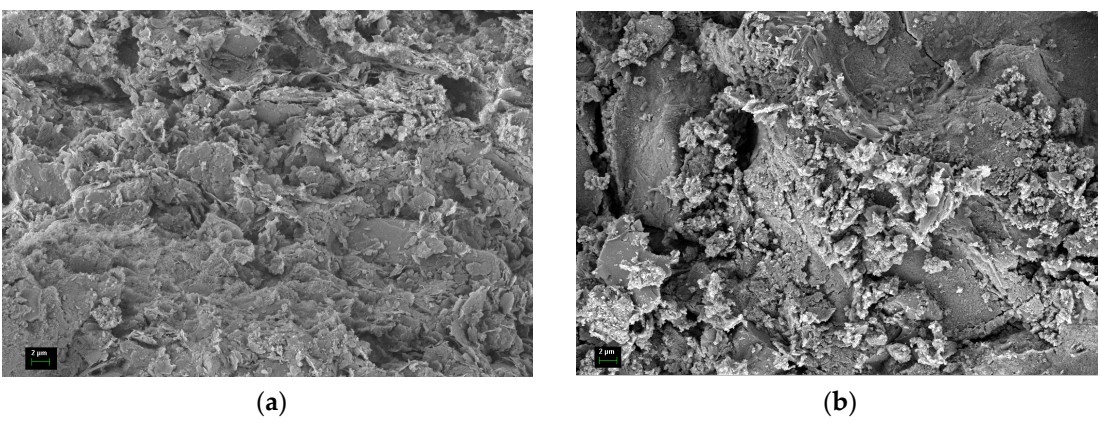

(**a**)                                          (**b**)

**Figure 5.** Superficial SEM pictures for (**a**) functional ceramsite and (**b**) bio-ceramsite.

In contrast to functional ceramsite without film formation, bio-ceramsite has dense microbial flocs on its surface that are mainly composed of bacillus. The bacteria are connected as a whole by EPS, mainly composed of proteins and polysaccharides, which are distributed in the rough parts of the ceramsite surface with more pores. These rough parts can supply a stable static hydraulic environment for microorganisms and prevent the biofilm structure from being damaged by water flow impact. However, there are still many pores in the flocculent microbial membrane even after the biofilm fully encases the ceramsite. Zhu et al. [32] characterized the surface morphology of the biofilm during the start-up of MBBR using atomic force microscopy, confirming the widespread existence of this pore structure. These pores on the surface of the biofilm serve an important role in the mass transfer process of nutrients, which is conducive to the adsorption of ammonium nitrogen in wastewater by bio-ceramsite [38].

*3.2. Static Adsorption of Ammonium Nitrogen*

3.2.1. Comparison between Fresh Functional Ceramsite and Bio-Ceramsite

Fresh functional ceramsite and bio-ceramsite were investigated experimentally to examine the impact of the biofilm on the ammonium nitrogen adsorption capacity of ceramsite, and the findings are shown in Figure 6.

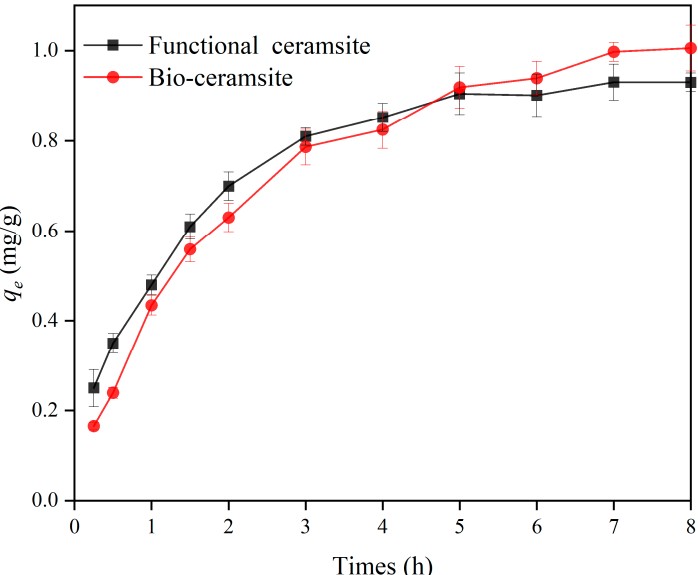

**Figure 6.** A comparison of ammonium adsorption capacity between bio-ceramsite and functional ceramsite.

As displayed in Figure 6, the adsorption equilibrium of bio-ceramsite and fresh ceramsite were achieved at about 7 h and 5 h, respectively. The $NH_4^+$-N adsorption capacity of bio-ceramsite was 1.0057 mg/g, and that of fresh ceramsite was 0.9305 mg/g. The above results showed that the existence of the biofilm affected the rate of ion exchange and increased the amount of time needed for bio-ceramsite to reach adsorption equilibrium. The biofilm, however, had less effect on the ceramsite ammonium nitrogen adsorption capacity. The findings were related to the pore and crack on the surface of bio-ceramsite observed by SEM (Figure 5). A large number of pores and cracks existed on the surface of bio-ceramsite which provided sufficient channels for ammonium ions to enter and exit bio-ceramsite without lowering its adsorption capacity. It is worth noting that the ammonium nitrogen adsorption capacity of bio-ceramsite was even slightly higher than that of fresh functional ceramsite, which may be because $Na^+$ from $NaHCO_3$ replaced the cations in bio-ceramsite zeolite components during the biofilm culturing process, completing the chemical modification of bio-ceramsite and increasing its ammonium nitrogen adsorption capacity [39].

3.2.2. Effect of Adsorbent Dosage

It is important to investigate the effect of dosage on the ammonium nitrogen adsorption of bio-ceramsite to optimize the usage of bio-ceramsite in practical applications and maximize its adsorption capacity. Therefore, the adsorption process was performed in the presence of 25 mg/L of $NH_4^+$-N solutions at 20 °C for 8 h, varying the adsorbent dose from 2 to 40 g/L. The results are demonstrated in Figure 7. The removal rate of ammonium nitrogen rose dramatically to 89.2% when the dosage of bio-ceramsite was increased to 28 g/L. However, with a further increase in dosage to 40 g/L, the removal rate of ammonium nitrogen increased gently from 89.2% to 92.1%. This is due to the fact that when the removal rate of ammonium nitrogen reached about 90%, the ammonium nitrogen in the solution had essentially been adsorbed, the free $NH_4^+$ was difficult to be adsorbed by bio-ceramsite. The unit ammonium nitrogen adsorption capacity of bio-ceramsite decreased slowly when the dosage of bio-ceramsite increased from 2 g/L to 10 g/L, and the adsorption capacity dropped sharply when the dosage of bio-ceramsite exceeded 10 g/L. The decline of adsorption capacity can be attributed to the adsorption competition between bio-ceramsite [40]. Bio-ceramsite had a higher removal rate of ammonium nitrogen when dosed at 10 g/L, which is closer to the saturation threshold.

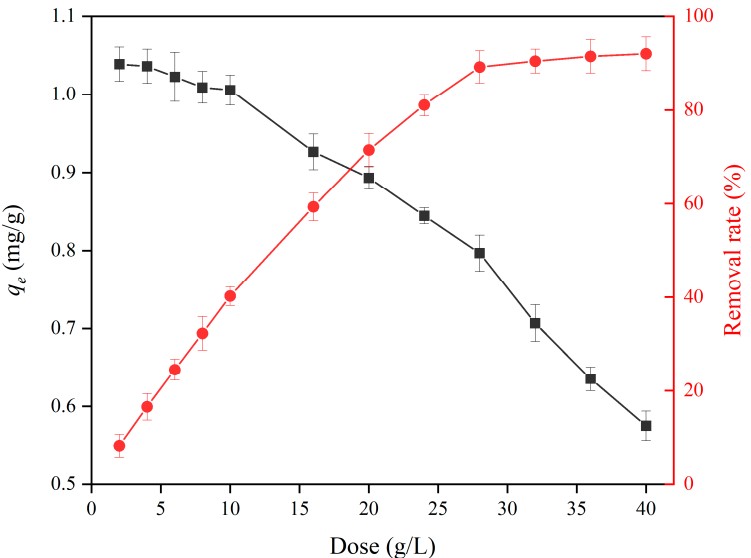

**Figure 7.** The effect of dosage on the adsorption capacity of bio-ceramsite.

### 3.2.3. Effect of pH

The pH value of wastewater will affect not only the existing form of ammonium nitrogen, but also the charge on the surface of the adsorbent, making the solution pH value an important factor in the ammonium nitrogen adsorption research [41]. The effects of pH on the adsorption of the $NH_4^+$-N were investigated by varying the pH from 4 to 11 at room temperature (20 °C) and keeping the initial $NH_4^+$-N concentration at 25 mg/L and the adsorbent dosage at 10 g/L. The results are illustrated in Figure 8. The bio-ceramsite adsorption capacity of ammonium nitrogen firstly increased, then decreased, and finally reached the maximum at pH 8. And the equilibrium between $NH_4^+$ and free ammonia in $NH_4Cl$ solution is observed as follows:

$$NH_4^+ + OH^- \rightleftharpoons NH_3 \cdot H_2O \tag{2}$$

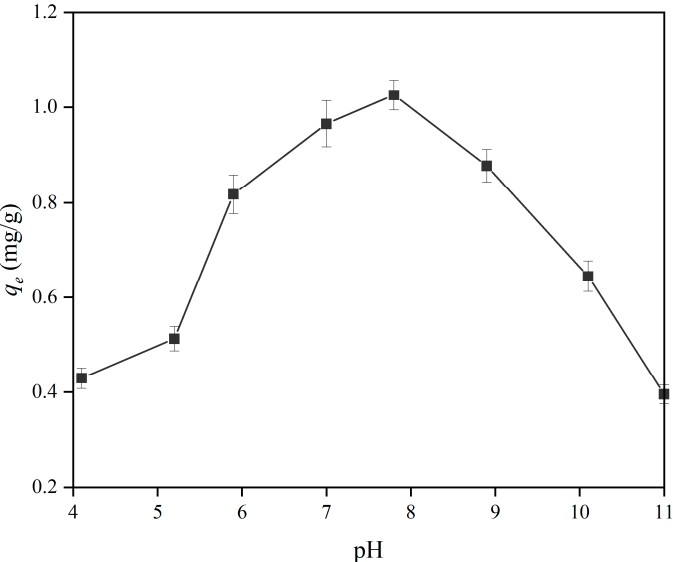

**Figure 8.** Effect of pH on the adsorption capacity of bio-ceramsite.

The minimal amount of adsorption at a low pH value can be attributed to the competition between $H^+$ and $NH_4^+$ for the bio-ceramsite surface [42]. The adsorption capacity

reached the maximum value when the pH of the solution was around 8.0. The morphology distribution of ammonium nitrogen in the solution gradually changed from $NH_4^+$ to $NH_3 \cdot H_2O$ when the pH rose [43]. Physical adsorption, rather than the ion exchange between zeolite components and $NH_4^+$ in ceramsite, predominated the adsorption of ammonium nitrogen on bio-ceramsite at this time. Therefore, the strong acid and strong alkali environment are not conducive to the adsorption of ammonium nitrogen by bio-ceramsite, while there is a range suitable for the adsorption of ammonium nitrogen by bio-ceramsite under weakly alkaline, neutral, and weakly acidic conditions, and the bio-ceramsite adsorption capacity of ammonium nitrogen does not change greatly in this range.

3.2.4. Effect of Ammonium Concentration

The adsorption capacity was determined as a function of $NH_4^+$-N concentration in the range of 10 mg/L to 300 mg/L by keeping the adsorbent dosage and duration fixed at 2.5 g and 8 h, respectively. An increase in adsorption capacity is shown (Figure 9) when $NH_4^+$-N concentrations rise. The adsorption capacity rose by almost 1.58 times from 0.89 mg/g to 1.60 mg/g when the $NH_4^+$-N concentration was raised from 10 mg/L to 300 mg/L. The huge $NH_4^+$-N concentration gradient between the adsorbent and the adsorbate is the key driving force in pushing up the adsorption capacity of bio-ceramsite in high-concentration $NH_4Cl$ solution [39].

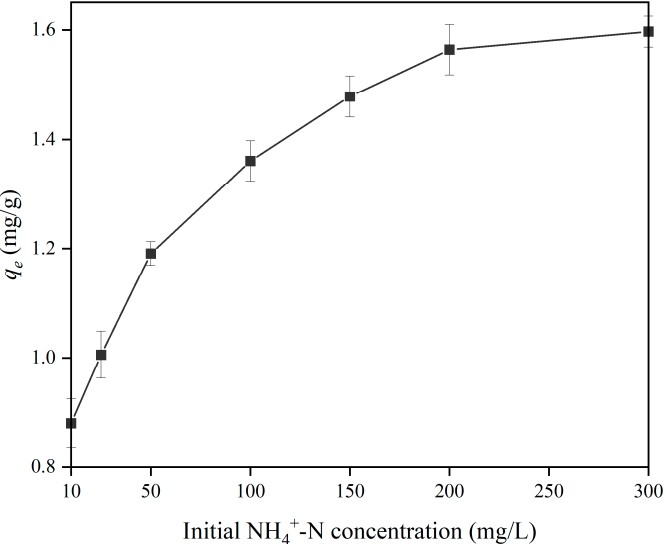

**Figure 9.** The effect of ammonium concentration on the adsorption capacity of bio-ceramsite.

*3.3. Mechanism of Ammonium Nitrogen Removal*

3.3.1. Adsorption Isotherm

The optimum efficiency of the adsorbents is determined in large part by adsorption isotherm, which refers to the interaction profile between the adsorbate and adsorbents. Therefore, the practical design and application of adsorption systems require the link between experimental data and theoretical or empirical equations. The experimental data of ammonium nitrogen was fitted to the most common adsorption isotherms developed by Langmuir–Freundlich.

(1)    Langmuir

Langmuir isotherm forecasts the monolayer adsorption on the surface of the adsorbate through a one–one interaction of the adsorbate with the active site of the adsorbent [44]. The relative parameters are calculated using Equation (3).

$$q_e = \frac{q_{Max} K_L C_e}{1 + K_L C_e} \tag{3}$$

where $q_e$ (mg/g) is the amount adsorbed at equilibrium; $C_e$ (mg/L) is equilibrium concentration; $q_{Max}$ is the theoretical maximum adsorption capacity of Langmuir; $K_L$ is the adsorption constants of Langmuir, L/mg.

The dimensionless constant separation factor $R_L$, given by Equation (4), is the essential characteristic of the Langmuir isotherm, with a value between 0 and 1 that envisages the spontaneity of the adsorption process [45].

$$R_L = \frac{1}{1 + K_L C_i} \tag{4}$$

(2)    Freundlich adsorption model

The Freundlich adsorption model is an empirical relation used to characterize complex interfacial systems and reversible adsorption systems, including, but not limited to, monolayer adsorption [46]. The adsorption data were fitted to Freundlich isotherm, represented by Equation (5).

$$q_e = K_F C_e^{\frac{1}{n}} \tag{5}$$

where $K_F$ and $1/n$ denote the adsorption capacity and intensity of adsorption, respectively; $K_F$ is the affinity coefficient of Freundlich adsorption isotherm model, (mg/g)/(mg/L); $1/n$ is the linear deviation degree of adsorption model, and is an infinite dimension constant. The intensity of the adsorption reaction can be determined by $1/n$. If $1/n$ is between 0.1 and 0.5, adsorption is easy, and the smaller the value of $1/n$, the stronger the adsorption capacity. If $1/n = 1$, linear adsorption occurs; if $1/n$ is greater than 2, adsorption is difficult. $K_F$ can measure the magnitude of the adsorbent adsorption capacity; the higher the value of $K_F$, the greater the adsorbent adsorption capacity.

The ammonium nitrogen adsorptive capacity of bio-ceramsite and the equilibrium concentration of ammonium nitrogen in the liquid phase can be characterized by the Langmuir and Freundlich adsorption isotherm equation, and the results are shown in Figure 10. The maximum adsorption capacity and adsorption mechanism of bio-ceramsite can be determined by analyzing the isothermal equation. Tables 4 and 5 exhibit the fitting parameters of the relevant adsorption isothermal model.

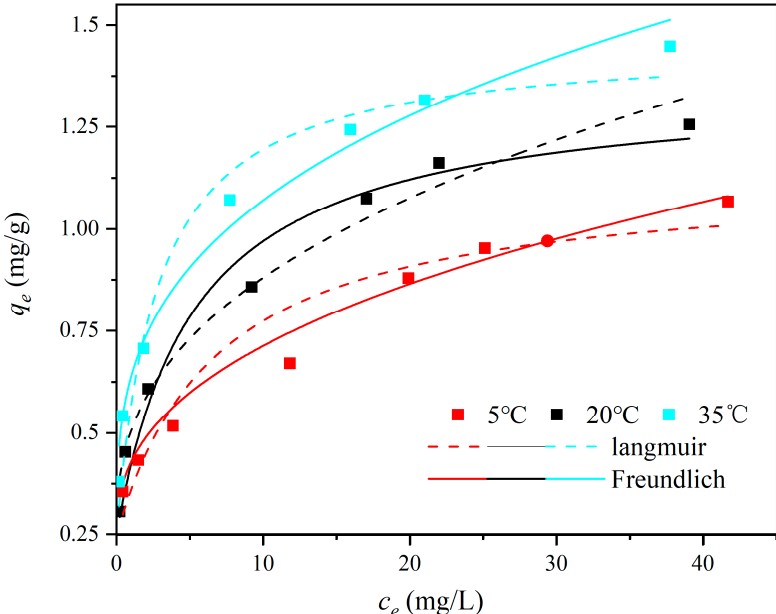

**Figure 10.** Adsorption isotherm models of bio-ceramsite at different temperatures.

**Table 4.** Parameters of adsorption isotherm models of bio-ceramsite at different temperatures.

| Temp (°C) | $q_{e, exp}$ (mg/g) | Langmuir Model | | | Freundlich Model | | |
|---|---|---|---|---|---|---|---|
| | | $q_{max, fitted}$ (mg/g) | $K_L$ (L/mg) | $R^2$ | $1/n$ | $K_F$ (mg/g)/(mg/L)$^{1/n}$ | $R^2$ |
| 5 | 1.0682 | 1.0094 | 0.1458 | 0.9523 | 0.3723 | 0.1809 | 0.9896 |
| 20 | 1.2557 | 1.3237 | 0.1883 | 0.9799 | 0.3911 | 0.2561 | 0.9804 |
| 35 | 1.4468 | 1.3723 | 0.3613 | 0.9771 | 0.3017 | 0.3888 | 0.9830 |

**Table 5.** Relation between the RL of bio-ceramsite and ammonium concentration.

| Temp (°C) | $C_0$ (mg/L) | | | | | | |
|---|---|---|---|---|---|---|---|
| | 5 | 8 | 10 | 15 | 25 | 35 | 50 |
| 5 | 0.5784 | 0.4068 | 0.2554 | 0.1206 | 0.0790 | 0.0642 | 0.0437 |
| 20 | 0.5151 | 0.3469 | 0.2098 | 0.0960 | 0.0623 | 0.0504 | 0.0342 |
| 35 | 0.3563 | 0.2168 | 0.1216 | 0.0525 | 0.0334 | 0.0269 | 0.0181 |

Figure 10 shows that the theoretical maximum adsorption capacity ($q_{max, fitted}$) and equilibrium adsorption capacity ($q_{e, exp}$) of bio-ceramsite increased with the rise in temperature, indicating that the adsorption of ammonium nitrogen is an endothermic process and temperature increase facilitates the development of the adsorption process [47]. As displayed in Table 4, the difference between $q_{e, exp}$, fitted and $q_{max, fitted}$ in the Langmuir isothermal adsorption model was at its smallest at 35 °C, implying that the increase in bio-ceramsite adsorption capacity became smaller with the increase in temperature, and bio-ceramsite approached the maximum theoretical adsorption capacity at 35 °C. The adsorption characteristics of the Langmuir isotherm model can be expressed by $R_L$, and the $R_L$ value was between 0 and 1 for all conditions, indicating that the $NH_4^+$-N adsorption of bio-ceramsite is preferential adsorption.

The $K_F$ value in the Freundlich isotherm model represents the adsorption capacity. It can be found by analyzing the fitting parameter data, so that the order of $K_F$ from large to small was also 35 °C > 20 °C > 5 °C, which is consistent with the conclusion of the Langmuir isotherm model. The value of $1/n$ ranges from 0 to 0.5 at different temperatures, which also indicates that the adsorption process of $NH_4^+$-N is preferential adsorption. By analyzing the corresponding regression correlation coefficient ($R^2$) values, it was discovered that the adsorption phenomena involving the ceramsite and the $NH_4^+$ ions in the present case closely follow the Freundlich isotherm model, in comparison to the Langmuir model. The acceptable fitting of Freundlich isotherms suggests that multilevel adsorbed $NH_4^+$ was formed during the reaction process [48].

### 3.3.2. Adsorption Kinetics

In order to further reveal the ammonium nitrogen adsorption mechanism, the experimental data were fitted to quasi-first order, quasi-second kinetic, and intra-particle diffusion models.

(1)  Quasi-first-order kinetic model

The quasi-first-order kinetic model was first proposed by Lagergren stating that time has a primary role in driving the change in adsorbents in the system [49]. The relative parameters of the quasi-first-order kinetic model can be calculated using Equation (6).

$$q_t = q_e\left(1 - e^{-k_1 t}\right) \tag{6}$$

where $q_e$ and $q_t$ stand for $NH_4^+$-N adsorbed per unit weight of adsorbent (mg/g) at equilibrium and at time t respectively; $k_1$ represents the quasi-first-order rate constant (min$^{-1}$).

(2)　Quasi-second-order kinetic model

Ho and McKay coined the quasi-second-order kinetic model through experiments, which incorporates basic adsorption processes, such as diffusion on the adsorbent surface, surface adsorption, and intra-particle diffusion [50]. The relative parameters of the quasi-second-order kinetic model can be calculated using Equation (7).

$$q_t = \frac{k_2 q_e^2 t}{1 + k_2 q_e t} \tag{7}$$

where $q_e$ and $q_t$ stand for $NH_4^+$-N adsorbed per unit weight of adsorbent (mg/g) at equilibrium and at time t, respectively; $k_2$ is the quasi-second-order rate constant, g/(mg min).

The nonlinear fitting between the $q_t$ at time t was analyzed to determine whether it is compliant with the quasi-first and quasi-second order kinetic models. The theoretical equilibrium adsorption capacities $q_{e1}$ and $q_{e2}$ were obtained by theoretical calculation under certain conditions.

(3)　Intra-particle diffusion

This model is often used to analyze the control steps in the reaction to determine the adsorption mechanism [51]. The relative parameters of the intra-particle diffusion kinetic model can be calculated using Equation (8).

$$q_t = k_s t^{\frac{1}{2}} + c \tag{8}$$

where $k_s$ represents the intra-particle diffusion model rate constant, mg/(g·min$^{1/2}$); and $c$ is the index of the boundary layer. The parameter $t^{\frac{1}{2}}$ was fitted using $q_t$ to identify the dominant step of adsorption. If there is no other rate-limiting step other than the diffusion process inside the particle, then the line passes through the origin.

The quasi-first-order and quasi-second-order kinetic fitting curves, and pertinent fitting parameters of the bio-ceramsite adsorption of ammonium nitrogen are shown in Figure 11 and Table 6.

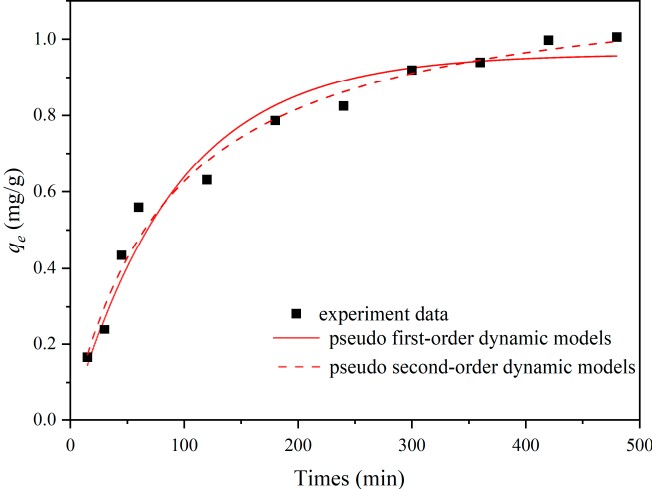

**Figure 11.** Quasi-first order, quasi-second order dynamic models for the ammonium adsorption on bio-ceramsite.

**Table 6.** Kinetic parameters for pseudo first-order and pseudo-second-order dynamic models for the adsorption of ammonium on bio-ceramsite.

| Dynamic Adsorption Model | Equations | $R^2$ | $q_{e, exp}$ (mg/g) | Parameters |
|---|---|---|---|---|
| first-order model | $y = q \times (1 - \exp(-k \times x))$ | 0.9559 | 1.0057 | $q_{e1}$ 0.9362 (mg/g) $k_1$ 0.0105 (min$^{-1}$) |
| second-order model | $y = k \times q^2 \times x/(1 + k \times q \times x)$ | 0.9701 | 1.0057 | $q_{e2}$ 1.1398 (mg/g) $k_2$ 0.0106 (g/(mg·min)) |

By analyzing the $R^2$ values, it can be observed that the experimental kinetic data of $NH_4^+$ fitted well to the second-order kinetic model, which yielded regression coefficients nearly equal to one ($R^2 = 0.9701$). The theoretical equilibrium adsorption capacity ($q_{e2}$) of 1.1398 mg/g may be due to the highly complex adsorption process of bio-ceramsite for ammonium nitrogen, including physical adsorption and chemical adsorption, van der Waals forces, electrostatic attraction, ion exchange, etc., while the quasi-second-order kinetic model can better characterize such a complex adsorption process in the solid–liquid system [52]. This intricate adsorption process is a combination of external membrane diffusion, surface adsorption, and internal diffusion of particles. However, the quasi-second-order adsorption kinetics model cannot determine the leading role of the adsorption process [53]. It is thus necessary to further elucidate the rate control mechanism of the adsorption process through the study of the internal particle diffusion model.

The internal particle diffusion kinetic model fitting curves and relevant fitting parameters of the adsorption of ammonium nitrogen by bio-ceramsite are displayed in Figure 12 and Table 7.

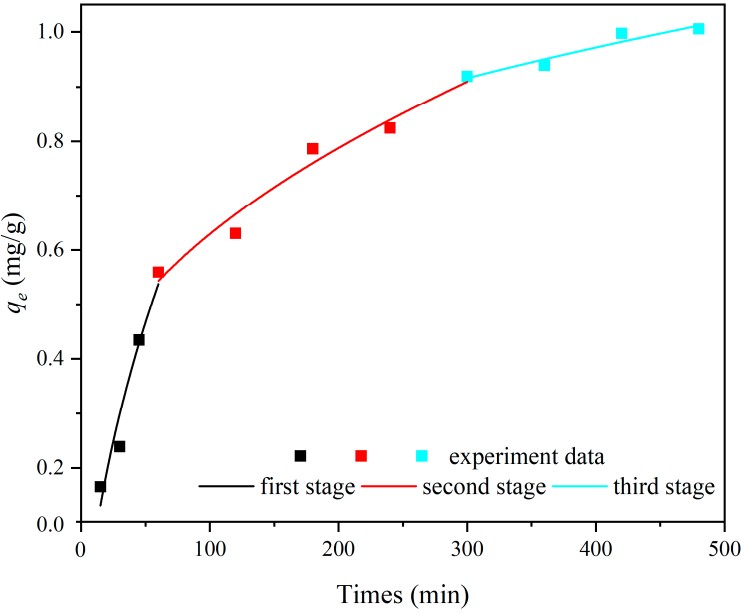

**Figure 12.** Intra-particle diffusion model for the adsorption of ammonium on bio-ceramsite.

**Table 7.** Kinetic parameters using the intra-particle diffusion model for the adsorption of ammonium on bio-ceramsite.

| Intra-Particle Diffusion Model | Equations | $k$ (mg/(g min$^{1/2}$)) | $c$ (mg/g) | $R^2$ |
|---|---|---|---|---|
| First stage (0~60 min) | $y = 0.1048 \times x^{1/2} - 0.2743$ | 0.1048 | −0.2743 | 0.9450 |
| Second stage (60~300 min) | $y = 0.0383 \times x^{1/2} + 0.2462$ | 0.0383 | 0.2462 | 0.9707 |
| Third stage (300~480 min) | $y = 0.0244 \times x^{1/2} + 0.4702$ | 0.0244 | 0.4702 | 0.6159 |

Figure 12 shows the three fitting curves at 0–60 min, 60–300 min, and 300–480 min respectively, corresponding to three different stages of the intra-particle diffusion model. None of the three fitting curves pass through the origin. The results demonstrated that the adsorption of $NH_4^+$ on bio-ceramsite did not occur by a process of single particle diffusion. The first step included the adsorbent $NH_4^+$ being diffused to the ceramsite surface through the biofilm. At this stage, there were more empty adsorption sites on the ceramsite surface, and the concentration gradient of ammonium nitrogen between the solution and the ceramsite surface was large, so the adsorption rate was fast. The adsorption capacity of ammonium nitrogen reached 0.5592 mg/g at the 60th min [54]. With the bio-ceramsite surface adsorption sites having decreased during the second stage, the adsorbate $NH_4^+$ reached monolayer saturation on the surface of ceramsite and progressively diffused into the internal pores of zeolite components in ceramsite (intra-particle diffusion). Due to the decrease in $NH_4^+$-N concentration, the increase in electrostatic repulsion and the narrowing of the zeolite component pore size in ceramsite, the adsorption process gradually slowed down. According to the existing adsorption kinetics theory, the adsorption velocity is dictated by the slower step in the reaction process, so the second stage served as the speed control step of the whole adsorption process [55]. The third stage included the dynamic equilibrium between adsorption and desorption. As the adsorption equilibrium would be disrupted by the continued oscillation, the fitting degree was not high with an $R^2$ of 0.6159.

### 3.3.3. Adsorption Thermodynamics

The energy changes, such as exothermic or endothermic, always undergo a reaction process in the adsorbent system. The nature and magnitude of thermodynamic parameters envisage the feasibility of the adsorption process and reveal the mechanism of the adsorbed layer formation [56]. Based on the ammonium nitrogen adsorption data of bio-ceramsite at different temperatures (278.15, 293.15, 308.15 K) and different concentrations (5, 8, 10, 15, 25, 35, 50 mg/L) in the isothermal adsorption test, the thermodynamic quantities, such as Gibbs free energy ($\Delta G^\theta$), enthalpy ($\Delta H^\theta$) and entropy ($\Delta S^\theta$), can be calculated via Equations (9)–(11):

$$K_C = \frac{C_0 - C_t}{C_t} \times \frac{V}{m} \tag{9}$$

$$\ln K_C = -\frac{\Delta H^\theta}{RT} + \frac{\Delta S^\theta}{R} \tag{10}$$

$$\Delta G^\theta = \Delta H^\theta - T\Delta S^\theta \tag{11}$$

where $C_0$ and $C_t$ are $NH_4^+$ concentration at the initial moment and at time t, respectively (mg/L); $V$ is the volume of the solution (mL); R is the gas constant, 8.314 J/(mol·K); T is the temperature in Kelvin (K); and $K_C$ represent the equilibrium constant between the adsorbed phase and the bulk aqueous solution (mL/g).

Thermodynamic curve (Figure 13) can be obtained by fitting $1/T$ with $lnK_C$, whose slope is $-\Delta H^{\theta}/R$ and intercept is $\Delta S^{\theta}/R$. $\Delta G^{\theta}$ can also be obtained according to the relationship between $\Delta G^{\theta}$, $\Delta H^{\theta}$, and $\Delta S^{\theta}$, and the results are shown in Table 8.

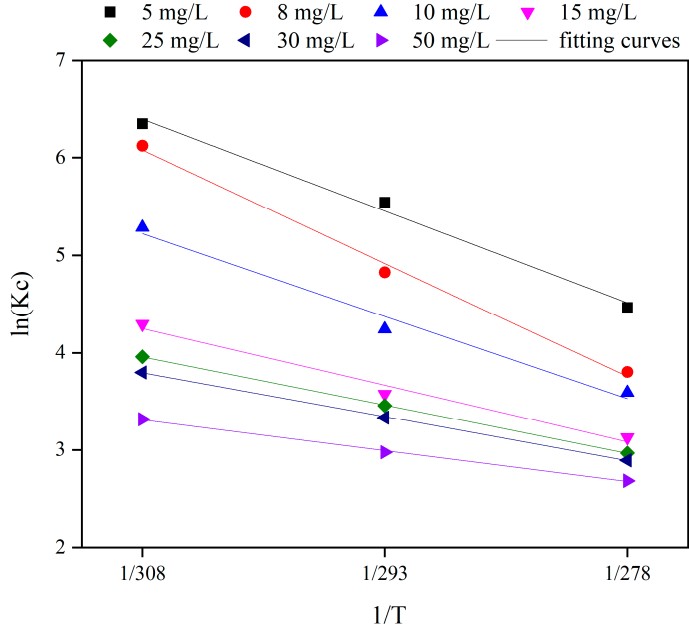

**Figure 13.** The relationship between $\Delta G^{\theta}$ and T for the adsorbed ammonium on bio-ceramsite under different reaction concentrations.

**Table 8.** Thermodynamic parameters for the adsorption of ammonium on bio-ceramsite.

| $C_0$ | T | $\Delta G^{\theta}$ | $\Delta H^{\theta}$ | $\Delta S^{\theta}$ | $R^2$ |
|---|---|---|---|---|---|
| mg/L | (K) | (kJ/mol) | (kJ/mol) | (kJ/mol) | |
| | 278.15 | −12.995 | | | |
| 5 | 293.15 | −15.123 | 26.4593 | 0.1419 | 0.9859 |
| | 308.15 | −17.250 | | | |
| | 278.15 | −11.449 | | | |
| 8 | 293.15 | 14.354 | 42.4238 | 0.1937 | 0.9910 |
| | 308.15 | −17.260 | | | |
| | 278.15 | −10.108 | | | |
| 10 | 293.15 | −12.293 | 30.4218 | 0.1457 | 0.9658 |
| | 308.15 | −14.479 | | | |
| | 278.15 | −8.386 | | | |
| 15 | 293.15 | −10.243 | 26.0594 | 0.12384 | 0.9632 |
| | 308.15 | −12.101 | | | |
| | 278.15 | −8.107 | | | |
| 25 | 293.15 | −9.416 | 16.1682 | 0.0873 | 0.9997 |
| | 308.15 | −10.725 | | | |
| | 278.15 | −7.827 | | | |
| 35 | 293.15 | −9.013 | 14.1762 | 0.0791 | 0.9994 |
| | 308.15 | −10.200 | | | |
| | 278.15 | −6.953 | | | |
| 50 | 293.15 | −7.945 | 11.4525 | 0.0662 | 0.9961 |
| | 308.15 | −8.938 | | | |

$\Delta G^{\theta}$ values were always negative under the conditions of different temperatures and different initial ammonium nitrogen concentrations, highlighting the spontaneity of the adsorption process. Furthermore, the decrease in $\Delta G^{\theta}$ with the increase in temperature suggested the process to be endothermic in nature [57]. The value of $\Delta H^{\theta}$ was greater than

zero, indicating the adsorption process of $NH_4^+$-N on bio-ceramsite was an endothermic process, and the temperature rise favored ammonium nitrogen adsorption. Otherwise, the value of $\Delta H^\theta$ ranged from 11.4525 to 42.4238 kJ/mol. The $\Delta H^\theta$ of physical adsorption is typically between 2.1 and 20.9 kJ/mol, while the $\Delta H^\theta$ of chemical adsorption is between 20.9 and 418.4 kJ/mol. Accordingly, the adsorption of $NH_4^+$-N by bio-ceramsite includes both physical and chemical adsorption [58]. The value of $\Delta S^\theta$ was consistently positive, suggesting that bio-ceramsite had an affinity for $NH_4^+$-N adsorption, which was in line with the results of the Freundlich adsorption isothermal model. With the increase in temperature, the active sites of bio-ceramsite increased, the confusion degree of the solid–liquid interface increased, and the reaction rate of removing ammonium nitrogen from wastewater to the bio-ceramsite accelerated.

## 4. Conclusions

Comparative analysis revealed that functional ceramsite has the advantage of bio-affinity. The presence of EPS facilitated the attachment and aggregation of microorganisms on the surface of functional ceramsite. The secretion of protein is positively associated with the ammonium nitrogen removal efficiency of MBBR. Results showed that bio-ceramsite had a slightly higher ammonium nitrogen adsorption capacity than fresh functional ceramsite. The biofilm has minimal effect on the adsorption capacity of ceramsite due to the existence of pores on its surface, but the biofilm lowered the ion exchange rate in ceramsite zeolite components. Additionally, the Freundlich adsorption isotherm model and the quasi-second-order kinetic model had better fitting effects on the $NH_4^+$-N adsorption process. The adsorption of bio-ceramsite to $NH_4^+$-N was an endothermic process that included physical and chemical adsorption. Furthermore, the results of adsorption thermodynamics suggested that bio-ceramsite has an affinity for the adsorption of $NH_4^+$-N. Through research, it was found that the studies remain in the superficial phase, and they fail in the practical applications of functional ceramsite. Numerous future studies are needed to enable the following: (1) the application of functional ceramsite in the MBBR adsorption–shortcut SND process, to compare it with the commercial ceramsite; (2) the desorption and regeneration characteristics of bio-ceramsite. In general, functional ceramsite has the potential to be a viable option for application in MBBR to enhance nitrogen removal from aquaculture wastewater.

**Author Contributions:** Conceptualization, L.W.; methodology, H.S.; software, L.W.; validation, N.Z. and H.S.; formal analysis, L.W. and X.M.; investigation, Q.W.; resources, X.S.; data curation, L.W.; writing—original draft preparation, L.W.; writing—review and editing, N.Z. and Y.A.H.; visualization, Q.W. and H.S.; supervision, N.Z.; project administration, L.W. and Y.A.H.; funding acquisition, X.S. and N.Z. All authors have read and agreed to the published version of the manuscript.

**Funding:** This study is supported by the Distinguished Postdoctoral Program of Jiangsu Province (2022ZB165), the Key Science and Technology Project of Water Resources Ministry (SKR—2022070), the National Key Research and Development Plan (2020YFD0900705), the Key Science and Technology Project for Nanjing Water Conservancy Bureau (2019—1) and the National Natural Science Foundation of China (42007018).

**Data Availability Statement:** The data presented in this study are available on request from the corresponding author.

**Acknowledgments:** The authors would like to acknowledge the help of Xiaohou Shao for the help of research guidance, and thank for Ningyuan Zhu for critical reading and improving an earlier version of this manuscript. Also, great thanks go to Qiling Wang in experimental data analysis.

**Conflicts of Interest:** The authors declare no conflict of interest.

## Abbreviations

Moving bed biofilm reactor (MBBR), simultaneous nitrification and denitrification (SND), dissolved oxygen (DO), free ammonia (FA), chemical oxygen demand (COD), total nitrogen (TN), specific oxygen uptake rate (SOUR), volatile solid content of the biofilm (VS), extracellular polymeric substances (EPS), extracellular polysaccharide (Po), extracellular protein (Pr), eoxyribonucleic acid (DNA), scanning electron microscopy (SEM).

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
