# Peer review of "The Effect of Functional Ceramsite in a Moving Bed Biofilm Reactor and Its Ammonium Nitrogen Adsorption Mechanism"

_water, doi:10.3390/w15071362_

Round 1

Reviewer 1 Report

Dear Authors,

the paper is written correctly. Among that I need to ask for small correction - please add information which exactly microscope was used during SEM.

Author Response

Point 1: the paper is written correctly. Among that I need to ask for small correction - please add information which exactly microscope was used during SEM.

Response 1: Thanks for your sincere suggestions. Scanning electron microscopy (SEM, ZEISS MERLIN compact-61-78) was used to characterize the bio-ceramsite and the functional ceramsite in order to figure out the morphological characteristics of biofilm on ceramsite.

Reviewer 2 Report

This work propose a functional ceramsite for ammonia nitrogen adsorption. In my opinion, it could be published but I didn’t found great conclusions. Authors must make effort “to sell” their product. I suggest to improve the discussion and conclusions to understand if Bio-ceramsite are better (or not) in front of Functional ceramsite.

One question in the experimental part (line 133). Is the pH important for you? What do you mean with sentence “pH was regulated between 6.5 and 8? I have not found the study with the pH value, so this wide range said that pH is whatever.

I have noticed also some edition errors:

-          Liters should be always with capital “L”. Please, revise mL in lines 158, 159 and 168

-          Figure 13 has an error in the Figure caption (it is said to be Figure 2)

-          Please, explain the meaning of NOB in line 72 (it is the first time that acronym appears)

Author Response

Response to Reviewer 2 Comments

Point 1: One question in the experimental part (line 133). Is the pH important for you? What do you mean with sentence “pH was regulated between 6.5 and 8? I have not found the study with the pH value, so this wide range said that pH is whatever.

Response 1: Thank you for your insightful comments. This is a mistake. It should be 7.5 to 8.0. According to previous studies, the pH affects the activities of various biochemical enzymes during microbial metabolism. The pH range for the growth of ammonium-oxidizing bacteria (AnAOB) and ammonia-oxidizing bacteria (AOB) is commonly 7.0 - 8.5, and the optimum value is 7.6 ±â€¯0.4 [1]. The pH value for nitrite-oxidizing bacteria (NOB) activity ranged from 6.0 to 7.5, while the pH suitable for denitrifying bacteria proliferation is about 7.0 - 8.5 [2]. When we inoculated the activated sludge sludge, pH regulated between 7.5 and 8.0 was beneficial to the the coexistence of an optimal microbial community, i.e., AOB, AnAOB, NOB and denitrifying bacteria for complete N removal.

Reference:

  1. Guo, Y.; Luo, Z.; Shen, J.; Li, Y.-Y. The main anammox-based processes, the involved microbes and the novel process concept from the application perspective. Frontiers of Environmental Science & Engineering 2021, 16, 84.
  2. Al-Hazmi, H.E.; Lu, X.; Grubba, D.; Majtacz, J.; Badawi, M.; MÄ…kinia, J. Sustainable nitrogen removal in anammox-mediated systems: Microbial metabolic pathways, operational conditions and mathematical modelling. Sci. Total Environ. 2023, 868, 161633.

Point 2: Liters should be always with capital “L”. Please, revise mL in lines 158, 159 and 168.

Response 2: We have revised ml to mL in this manuscript.

Point 3: Figure 13 has an error in the Figure caption (it is said to be Figure 2).

Response 3: The mistake has been corrected.

Point 4: Please, explain the meaning of NOB in line 72 (it is the first time that acronym appears).

Response 4: NOB only appears once in this manuscript. We have supplied the complete spelling of NOB (nitrite-oxidizing bacteria).

Reviewer 3 Report

The article is very interesting, and the authors have done a laborious experimental work. The results obtained are conclusive and have been demonstrated in drawing conclusions.

However, in my opinion there are some aspects to improve the article:

 For the best follow-up of the article, it would be interesting to introduce a glossary of acronyms.

Line 103.  Please, change the points (Shift+3), there is a lot of disproportion in the water molecule of the chemical formulation. Example: in Table 1. Components of synthetic wastewater, also in the 105 line Na8Al8Si40O96•24H2O, etc.

Line 114. In Figure 1. Schematic diagram of the experimental facility, please fill the arrows indicating the flow direction.

Line 125 and following (195 ec. 1). The authors refer to Oxygen Demand (DO) but they must indicate that it is Biological Oxygen Demand (BOD) or COD (Chemical Oxygen Demand) at least in the text it must be specified, it could be confused.

Line 147: The number of experiences carried out is quite high, and this enriches the article, perhaps for future occasions the authors could carry out a Design of Experiments (DOE), although this is simply a recommendation.

Line 224: TN = total nitrogen please include it in the text.

Line 294. The authors indicate a color change of the biofilm inside and outside of the ceramsite. Indicating that the color of the biofilm on the surface was yellowish brown and the color of the biofilm inside the ceramsite pores was blackish gray.

Perhaps a microphotograph taken with a magnifying glass could be added to show this color change. IF it´s not possible in this publication for future publications this effect would be interesting to see.

Line 383 and following. Please, remove bold in NH4 + in the section title 3.2.4., and in the figure 9 and 11 title, in the tables  5 to 8, etc.

Line 501. In Table 6. Kinetic parameters for pseudo first-order, pseudo-second-order dynamic models, please move the units of k1 to the same line of the variable     k1 0.0105 (min-1).

 Line 561: Please review the numbering of the figures and tables, for example The Figure 2. The relationship between ∆?? and T for the adsorbed ???+-N on bio-ceramsite…… is the figure number 13.

 Line 581. The conclusions of the document should be more extensive. Many of the conclusions have been obtained in the different sections and could simply be summarized in the conclusions of the article.

Author Response

Response to Reviewer 3 Comments

Point 1: For the best follow-up of the article, it would be interesting to introduce a glossary of acronyms.

Response 1: Thank you very much for your suggestions. Glossary of acronyms has been introduced at the beginning of the article, i.e., moving bed biofilm reactor (MBBR), simultaneous nitrification and denitrification (SND), dissolved oxygen (DO), free ammonia (FA), chemical oxygen demand (COD), total nitrogen (TN), specific oxygen uptake rate (SOUR), volatile solid content of biofilm (VS), extracellular polymeric substances (EPS), extracellular polysaccharide (Po), extracellular protein (Pr), eoxyribonucleic acid (DNA), scanning electron microscopy (SEM).

Point 2: Line 103. Please, change the points (Shift+3), there is a lot of disproportion in the water molecule of the chemical formulation. Example: in Table 1. Components of synthetic wastewater, also in the 105 line Na8Al8Si40O96•24H2O, etc.

Response 2: Thank you for your insightful comments. We have modified the molecular formula of the related chemical formulation, i.e., CaCl2•2H2O, MnCl2•4H2O, and deleted Na8Al8Si40O96•24H2O.

Point 3: Line 114. In Figure 1. Schematic diagram of the experimental facility, please fill the arrows indicating the flow direction.

Response 3: We have indicated the flow direction in the schematic diagram of the experimental facility.

Figure 1. Schematic diagram of the experimental facility.

Point 4: Line 125 and following (195 ec. 1). The authors refer to Oxygen Demand (DO) but they must indicate that it is Biological Oxygen Demand (BOD) or COD (Chemical Oxygen Demand) at least in the text it must be specified, it could be confused.

Response 4: DO means dissolved oxygen. And we have indicated the Oxygen Demand is Chemical Oxygen Demand (COD).

Point 5: Line 147: The number of experiences carried out is quite high, and this enriches the article, perhaps for future occasions the authors could carry out a Design of Experiments (DOE), although this is simply a recommendation.

Response 5: We quite agree with your suggestion. The conclusions drawn from design of experiments may be more reliable. In our future research, we must carry out a Design of Experiments.

Point 6: Line 224: TN = total nitrogen please include it in the text.

Response 6: The complete spelling of TN has been added.

Point 7: Line 294. The authors indicate a color change of the biofilm inside and outside of the ceramsite. Indicating that the color of the biofilm on the surface was yellowish brown and the color of the biofilm inside the ceramsite pores was blackish gray. Perhaps a microphotograph taken with a magnifying glass could be added to show this color change. IF it´s not possible in this publication for future publications this effect would be interesting to see.

Response 7: Thank you very much for your suggestions. We agree that a microphotograph taken with a magnifying glass could show this color change better. But with the end of the experiment, it's hard to retake photos. In future experiments, we must pay attention to take microphotograph.

Point 8: Line 383 and following. Please, remove bold in NH4+-N in the section title 3.2.4., and in the figure 9 and 11 title, in the tables  5 to 8, etc.

Response 8: The bold in ammonium have been removed.

Point 9: Line 501. In Table 6. Kinetic parameters for pseudo first-order, pseudo-second-order dynamic models, please move the units of k1 to the same line of the variable     k1 0.0105 (min-1).

Response 9: The unit of k1 has been removed to the same line of the variable k1 0.0105 (min-1).

Point 10: Line 561: Please review the numbering of the figures and tables, for example The Figure 2. The relationship between ∆?? and T for the adsorbed -N on bio-ceramsite…… is the figure number 13.

Response 10: Thank you for your insightful comments. The numbering of the figures and tables have been reviewed.

Point 11: Line 581. The conclusions of the document should be more extensive. Many of the conclusions have been obtained in the different sections and could simply be summarized in the conclusions of the article.

Response 11: Comparative analysis revealed that the functional ceramsite has the advantage of bio-affinity. The presence of EPS facilitated the attachment and aggregation of microorganisms on the surface of functional ceramsite. The secretion of protein is positively associated with the ammonium nitrogen removal efficiency of MBBR. Results showed that the bio-ceramsite had a slightly higher ammonium nitrogen adsorption capacity than fresh functional ceramsite. The biofilm has minimal effect on the adsorption capacity of ceramsite due to the existence of pores on its surface, but the biofilm lowered the ion exchange rate in ceramsite zeolite components. Additionally, Freundlich adsorption isotherm model and the quasi-second-order kinetic model had better fitting effects on the ammonium adsorption process. The adsorption of bio-ceramsite to ammonium was an endothermic process including physical and chemical adsorption. And the results of adsorption thermodynamics suggested that the bio-ceramsite has an affinity for the adsorption of ammonium. Through research, it was found that the studies remain the superficial phase, and they fail to practical applications of functional ceramsite. Many Future studies are needed to do: 1) the applications of functional ceramsite in MBBR adsorption–shortcut SND process and compared with the commercial ceramsite; 2) desorption and regeneration characteristics of bio-ceramsite. In general, the functional ceramsite has the potential to be a viable option for application in MBBR to enhance nitrogen removal from aquaculture wastewater.

Reviewer 4 Report

This manuscript examines the performance of a ceramsite-based biofilm reactor in removing ammonium nitrogen from wastewater. Overall, it reads well and results are interesting. It can be published after the following comments are addressed:

Line 99, it should be ammonium nitrogen rather than ammonia nitrogen (which is only present at high pH). Please correct the rest.

Figure 1. the use of removal rate is not very accurate. Removal rate is often used to indicate how fast a contaminant is removed rather than percentage removal. Removal efficiency may be a more suitable term. The legends for the squares and triangles should be NH4+-N removal and TN removal.

Line 244, what does change law mean?

Figure 4. because protein has N in it, the increasing proportion of protein relative to polysaccharide may have something to do with increasing ammonium removal shown in Figure 2?

Figure 6 and Section 3.3, how did you differentiate between adsorption and biodegradation when using bio-ceramsite? 

Line 584, can you explain why a Pr/Po value of 3.7 means biofilm maturity. 

Author Response

Response to Reviewer 4 Comments

Point 1: Line 99, it should be ammonium nitrogen rather than ammonia nitrogen (which is only present at high pH). Please correct the rest.

Response 1: Thank you for your insightful comments. All of ammonia nitrogen has been replaced by ammonium nitrogen.

Point 2: Figure 1. the use of removal rate is not very accurate. Removal rate is often used to indicate how fast a contaminant is removed rather than percentage removal. Removal efficiency may be a more suitable term. The legends for the squares and triangles should be NH4+-N removal and TN removal.

Response 2: Thanks for your sincere suggestions. All of the removal rate has been replaced by removal efficiency. And the legends have also been modified, as shown in Figure 1.

Figure 2. The removal efficiency of ammonium and TN during the start-up period of MBBR.

Point 3: Line 244, what does change law mean?

Response 3: Replacing “change law” to “change rules” maybe easier to understand.

Point 4: Figure 4. because protein has N in it, the increasing proportion of protein relative to polysaccharide may have something to do with increasing ammonium removal shown in Figure 2?

Response 4: The proportion of Pr in the biofilm predominated over that of Po, and these differences vary greatly across stages. Part of ammonium nitrogen in the wastewater were assimilated by microbes to make proteins, which lead to the increase of protein proportion relative to polysaccharide and the reduce of ammonium nitrogen [1]. Shi et al. [2] believed that the change of protein content in EPS reflected the ability of microbial cells to secrete extracellular enzymes that were primarily involved in the capture and transportation of pollutants. In general, the secretion of protein is positively associated with the ammonium nitrogen removal efficiency of MBBR.

Reference:

  1. Zhang, J.; Miao, Y.; Zhang, Q.; Sun, Y.; Wu, L.; Peng, Y. Mechanism of stable sewage nitrogen removal in a partial nitrification-anammox biofilm system at low temperatures: Microbial community and EPS analysis. Bioresour. Technol. 2020, 297, 122459.
  2. Shi, Y.; Liu, Y. Evolution of extracellular polymeric substances (EPS) in aerobic sludge granulation: Composition, adherence and viscoelastic properties. Chemosphere 2021, 262, 128033.

Point 5: Figure 6 and Section 3.3, how did you differentiate between adsorption and biodegradation when using bio-ceramsite?

Response 5: The bio-ceramsite would be used in MBBR adsorption–shortcut SND process. The MBBR adsorption–shortcut SND process can be divided into two parts: the adsorption stage and the SND denitrification and regeneration stage. In the adsorption stage, the ammonium in wastewater firstly diffused to the ceramsite surface through the biofilm. Referring to the primary an ammonia nitrogen discharge standard in cities sewage treatment plant pollutant discharged standard (GB 18918-2002), 5 mg/L was chosen as the adsorption breakthrough point of ceramsite. When the ammonia nitrogen discharge concentration exceeded 5 mg/L, the influent and effluent ceased, and the MBBR entered the shortcut SND denitrification and regeneration stage. At this time, the adsorbed ammonia nitrogen would be desorbed by biochemical regeneration effect. Subsequently, the bio-ceramsite will recover its ammonium nitrogen adsorption capacity. More details about the MBBR adsorption–shortcut SND process, especially the optimal operating parameters in the adsorption and regeneration stage, would be show in our subsequent articles.

Point 6: explain why a Pr/Po value of 3.7 means biofilm maturity.

Response 6: Numerous studies have revealed that the change of the Pr/Po ratio can serve as a characteristic index of biofilm growth and shedding, and the Pr/Po value of biofilm in a stable reactor is around 3.38-4.57 [3]. After day 21, the value of Pr/Po in this test was stable at about 3.70, just between 3.38 and 4.57, indicating that the biofilm on the ceramsite was mature and the reactor’s biofilm formation had been successfully initiated.

Reference:

  1. Zhu, Y.; Zhang, Y.; Ren, H.; Geng, J.; Xu, K.; Huang, H.; Ding, L. Physicochemical characteristics and microbial community evolution of biofilms during the start-up period in a moving bed biofilm reactor. Bioresour. Technol. 2015, 180, 345-351.

Round 2

Reviewer 4 Report

The authors have addressed all the comments.